

# Chemical synthesis of peptidoglycan mimetic–disaccharide-tetrapeptide conjugate and its hydrolysis by bacteriophage T5, RB43 and RB49 L-alanyl-D-glutamate peptidases

Viatcheslav Azev[1], Alexey Chulin[1], Maxim Molchanov[2], Dmitry Prokhorov[2], Galina Mikoulinskaia[1], Vladimir N. Uversky[3,4] and Viktor Kutyshenko[2]

[1] Branch of Shemyakin and Ovchinnikov's Institute of Bioorganic Chemistry, Russian Academy of Sciences, Pushchino, Moscow Region, Russia
[2] Institute for Theoretical and Experimental Biophysics, Russian Academy of Sciences, Pushchino, Moscow Region, Russia
[3] Laboratory of New Methods in Biology, Institute for Biological Instrumentation of the Russian Academy of Sciences, Federal Research Center "Pushchino Scientific Center for Biological Research of the Russian Academy of Sciences", Pushchino, Moscow Region, Russia
[4] Department of Molecular Medicine and USF Health Byrd Alzheimer's Research Institute, Morsani College of Medicine, University of South Florida, Tampa, Florida, United States

Corresponding authors
Vladimir N. Uversky,
vuversky@usf.edu
Viktor Kutyshenko,
kutyshenko@rambler.ru

## ABSTRACT

**Background:** Endolysins of a number of bacteriophages, including coliphages T5, RB43, and RB49, target the peptidoglycans of the bacterial cell wall. The backbone of these bacterial peptidoglycans consist of alternating N-acetylglucosamine and N-acetylmuramic acid residues that is further "reinforced" by the peptide subunits. Because of the mesh-like structure and insolubility of peptidoglycans, the processes of the peptidoglycan binding and hydrolysis by enzymes cannot be studied by spectral methods. To overcome these issues we synthesized and analyzed here one of the simplest water soluble peptidoglycan mimetics.

**Methods:** A compound has been synthesized that mimics the peptidoglycan fragment of the bacterial cell wall, N-acetylglucosaminyl-β(1-4)-N-acetylmuramoyl-l-alanyl-γ-d-glutamyl-l-alanyl-d-alanine. NMR was used to study the degradation of this peptidoglycan mimetic by lytic l-alanoyl-d-glutamate peptidases of colibacteriophages T5, RB43, and RB49 (EndoT5, EndoRB43, and EndoRB49, respectively).

**Results:** The resulting glycopeptide mimetic was shown to interact with the studied enzymes. Its hydrolysis occurred through the bond between l-Ala and d-Glu. This artificial substrate mimetic was hydrolyzed by enzymes at different rates, which decreased outside the pH optimum. The EndoT5 demonstrated the lowest hydrolysis rate, whereas the EndoRB49-driven hydrolysis was the fastest one, and EndoRB43 displayed an intermediate potency. These observations are consistent with the hypothesis that EndoRB49 is characterized by the lowest selectivity, and hence the potentially broader spectrum of the peptidoglycan types subjected to hydrolysis, which was put forward in the previous study. We also show that to hydrolyze this

glycopeptide mimetic, enzymes approach the glycopeptide near the methyl groups of all three alanines.

# INTRODUCTION

For the release of phage progeny from host bacterial cell, lytic bacteriophages induce the synthesis of enzymes capable of degrading peptidoglycan of the bacterial cell wall either by hydrolysis of the peptide or glycosidic bonds, or by the non-hydrolytic rupture of the latter (*Young, 2014*). Bacterial peptidoglycan is a complex polymeric structure. It has a carbohydrate backbone consisting of alternating N-acetylglucosamine and N-acetylmuramic acid residues linked by β(1-4) glycosidic bonds that is further "reinforced" by the peptide subunits, often linked by interpeptide bridges. The stem peptide is linked to the muramic acid residue by an amide bond. The composition of the peptide subunit, as well as the structure of peptide bridges, vary, and these variations underlie the classification of peptidoglycans (*Schleifer & Kandler, 1972*). Types A and B differ in the branching position (from third amino acid residue of one stem peptide to fourth of the other, or from second to fourth, respectively) and the amino acid at the N-terminus of the stem peptide (for type A peptidoglycan, this is l-Ala). Next in the nomenclature is a number that reflects the structure of the bridge (for example, one corresponds to direct cross-linking between peptides stems, with no bridging amino acids). Finally, the Greek letter in the nomenclature reflects the type of amino acid at position 3 (e.g., γ-diaminopimelic acid, DAP). The amino acid at position 3 is characterized by the greatest variability: l-Lys, l-Orn, DAP are often found, less often meso-Lanthionine, l-5-Hydroxylysine, l-Ala, l-Glu, and others are found (*Vollmer, Blanot & de Pedro et al., 2008*). In peptidoglycans of type A, which are characteristic of most Gram-positive and all Gram-negative bacteria, the l-Ala-d-Glu bond is always present. This bond is the target of endolysins of a number of bacteriophages, including coliphages T5, RB43, and RB49 (*Mikoulinskaia et al., 2018*; *Mikoulinskaia et al., 2013*).

Structural methods (most often solid-state NMR) are used to study peptidoglycans and their structures (*Bougault et al., 2019*; *Kern et al., 2008*). Structural studies of enzymes for which peptidoglycan is a substrate are more complicated. To analyze the binding and hydrolysis of such a complex substrate by enzymes, usually soluble peptidoglycan (PG) fragments obtained as a result of partial hydrolysis by other enzymes (for example, muramidase or lysostaphin) are used (*Maya-Martinez et al., 2018*). However, the hydrolysates obtained in this way are difficult to standardize. Unconditional unification is achieved by using synthetic compounds that mimic a natural substrate, and such studies are known primarily for enzymes of Gram-positive bacteria (*Lee et al., 2019*; *Perez-Dorado et al., 2007*). In this case, both disaccharide dipeptides (muramyl dipeptides (MDPs) or glucosaminylmuramyl dipeptides (GMDPs)) and larger molecules are used.

The natural diversity of amino acids in position 3 of the PG peptide subunit led us to the idea of creating a mimetic that does not completely repeat the structure of natural peptidoglycan, but mimics its fragment as much as possible. This mimetic can be used to study the mechanism of binding and hydrolysis of peptidoglycan by L-alanyl-D-glutamate peptidases, which have high specific activity on the natural substrate. The carbohydrate moiety in such a mimetic should have been represented by a minimal disaccharide. The peptide is naturally long: it consists of four amino acids, the last of which is d-alanine. The choice of one more l-alanine in the 3 position of the peptide is due to the following considerations. First, a study of the interaction of the long-known peptidoglycan mimetic GMDP with our enzymes showed that this compound is weakly fixed by the protein in the l-alanine region, but hydrolysis of the peptide bond does not occur. In peptidoglycan of the type A1γ in the 3 position after l-Ala-d-Glu there is meso-diaminopimelic acid (DAP), which contains a rather extended hydrophobic region-$(CH_2)_3$, followed by d-alanine. We hypothesized that hydrophobic interactions play an important role in peptide binding and, possibly, it is their number that is lacking in GMDP. A bridging of two alanines was chosen to provide sufficient hydrophobicity and ease of identification by NMR. The entire construction of the disaccharide-tetrapeptide as a whole was designed as a basic one, in which it is planned to introduce further directed variations to simulate experiments aimed at studying various enzymatic properties (reaction rate, processivity, affinity for the substrate).

In short, it is impossible to study the process of the substrate binding and hydrolysis by enzymes by spectral methods, which provide detailed structural information, since natural peptidoglycans are insoluble. In this work, an attempt was made to circumvent these limitations by synthesizing one of the simplest water soluble peptidoglycan mimetics and to use this compound to study the process of its hydrolysis in vitro by recombinant L-alanyl-D-glutamate peptidases of bacteriophages T5, RB43, and RB49 (EndoT5, EndoRB43, and EndoRB49).

## MATERIALS & METHODS

All solvents (except HPLC grade acetonitrile), N-methylmorpholine, pyridine, trifluoroacetic acid were purified before the use according to the standard procedures (*Armarego & Chai, 2012*). Dimethylformamide, N-methylmorpholine, and pyridine were kept over four A molecular sieves after purification. Dichloromethane was kept over barium oxide. d6-DMSO and $D_2O$ were supplied by Cambridge Isotope Laboratories (USA). N-Acetylglucosaminyl-β(1-4)-N-acetylmuramic acid was a generous gift of Dr. Vasiliy N. Stepanenko (Experimental Biotechnological Manufacture, IBCh RAS). Bis (pentafluorophenyl) carbonate was prepared as described elsewhere (*Medvedkin et al., 1989*).

TLC was performed on Merck F254 silica gel G plates (part 1.05554.0001). Spots were detected by (1) dipping the eluted plates into the ninhydrin solution (ninhydrin 0.5 g, butanol-1 250 mL, acetic acid 50 mL, *sym*-collidine 10 mL) followed by heating with a heat-gun; (2) dipping the eluted plates into the phosphomolybdic acid solution (phosphomolybdic acid 6 g, ethanol 70 mL) followed by heating with a heat-gun.

NMR spectra were acquired on Bruker AVANCE III spectrometer (Bruker BioSpin, Germany) ($^1$H at 600 MHz, $^{13}$C at 125 MHz) and were referenced to a residual solvent signals. Unless otherwise stated, the spectra were taken at sample concentrations *ca.* 7–14 mg/mL in standard 5 mm NMR ampules using standard pulse sequences. Peak positions are reported in ppm, coupling constants are reported in Hertz (Hz). High-resolution mass-spectra were acquired on Orbitrap Elite Hybrid Ion Trap-Orbitrap Mass Spectrometer.

HPLC monitoring was accomplished on an instrument equipped with Waters 1525 binary HPLC pump and Waters 2487 Dual λ Absorbance Detector using a Phenomenex Synergi Hydro-RP C18 column (0.46 cm × 25 cm) and UV detection at 226 nm eluting at 1 mL min−1. Solvent A: water (0.1% TFA), solvent B: acetonitrile. Gradient 1: 0–1 min: 0% B; 1–20 min: 0–55% B; 20–21 min: 55–0% B; 21–25 min: 0% B. Gradient 2: 0–2 min: 3% B; 2–18 min: 3–50% B; 18–19 min: 50–3% B; 19–25 min: 3% B.

### *N*α-tert-butoxycarbonyl-l-alanyl-α-tert-butyl-d-glutamyl-l-alanyl-d-alanine tert butyl ester (4)

Compound **2** (570 mg, 1.5 mmol) was dissolved in DCM (10 mL) with stirring in 25 mL round bottom flask. Pentafluorophenyl trifluoroacetate (345 µL, 2 mmol) and pyridine (175 µL, 2.2 mmol) were successively added to the substrate solution, the flask was tightly closed with a glass stopper and the reaction mixture was stirred overnight. The reaction mixture was diluted with DCM (20 mL) and the resulting solution was successively washed with 0.1M aqueous NaHCO$_3$ (15 mL), 0.1 M aqueous HCl (15 mL) and water (20 mL). The organic layer was dried with MgSO$_4$ and evaporated. The resulting oily product was dissolved in ethyl acetate (2 mL) and the solution was slowly diluted with hexane (up to 15 mL). The precipitated activated ester was filtered off, washed with hexane (210 mL) and dried in dessicator over phosphorus pentoxide and paraffin to provide 412 mg of white solid that was homogeneous by TLC (Rf = 0.54, silica gel, 1:1 v/v EtOAc/n-heptane).

A part of the solid obtained (362 mg, 0.67 mmol) was dissolved in MeCN (5 ml) with stirring. Compound **3** (185 mg, 0.67 mmol) followed by NMM (150 µL, 1.34 mmol) were added to the solution. The reaction mixture was stirred at rt and in *ca.* 10 min precipitate formation was observed. DCM (10 mL) was added to the suspension and the reaction mixture was stirred overnight. The reaction mixture was filtered, the solid material on the filter was rinsed with DCM (3.2 mL), the filtrates were combined and evaporated in vacuo. EtOAc (25 mL) followed by water (25 mL) were added to the residue obtained. The aqueous layer was separated and the organic layer was washed with 0.1 M aqueous NaHCO$_3$ (10 mL), 0.1M aqueous HCl (10 mL) and water (20 mL). The organic layer was dried with Na$_2$SO$_4$, filtered and evaporated. Ether (10 mL) was added to the oily residue and the reaction product solidified. The solid obtained was filtered, washed with ether (25 mL) and dried in vacuo overnight to provide **4** (239 mg, 62%). For compound **4**: m.p. 175–176 ; Rf = 0.16 (silica gel, 9:1 v/v CHCl$_3$/MeOH), Rf = 0.74 (silica gel, 8:1:1 v/v/v MeCN/CHCl$_3$/AcOH); $^1$H NMR (DMSO-d6) δ 1.14–1.21 (m, 6H, CH$_3$-Ala), 1.23 (d, *J* = 7.32 Hz, 3H, CH$_3$-D-Ala), 1.29–1.46 (m, 27H, Boc, 2tBu),

1.69–1.82 (m, 1H, CHβGlu), 1.87–1.98 (m, 1H, CHβGlu), 2.14 (ψt, $J$ = 7.97 Hz, 2H, CHγGlu), 3.83–4.15 (m, 3H, CHα-D-Ala, CHα-Ala, CHαGlu), 4.29 (dq, $J$ = 7.09, 7.53 Hz, 1H, CHα-Ala), 6.82 (d, $J$ = 7.27 Hz, 1H, NHBoc), 7.96 (d, $J$ = 7.53 Hz, 1H, NHAla), 8.02 (d, $J$ = 7.99 Hz, 1H, NH-D-Glu), 8.10 (d, $J$ = 7.31 Hz, 1H, NH-D-Ala); [13]C NMR δ 172.82, 172.08, 171.56, 170.90, 170.81, 154.94, 80.58, 80.29, 78.03, 52.14, 49.63, 48.23, 47.94, 31.31, 28.14, 27.56, 27.11, 18.36, 18.30, 17.01; HRMS (+ESI) $m/z$ for $C_{27}H_{49}N_4O_9$ [M + H]$^+$ calculated 573.3494, observed: 573.3499.

### l-alanyl-γ-d-glutamyl-l-alanyl-d-alanine trifluoroacetate (5)

Compound **4** (239 mg, 0.41 mmol) was dissolved in a mixture of trifluoroacetic acid and water (95:5 v/v, 4 mL) with stirring. The reaction mixture was kept at 20 for 60 min and evaporated. Ether (7 mL) was added to the evaporation residue and the solid material obtained was separated from the ether mother liquor using centrifuge. The solid was resuspended and centrifuged in ether (3 × 1.5 mL) before it was dried in vacuo to provide **5** (181 mg, 92 %) that was used directly in the next step without further purification. For compound **5**: m.p. 139-140; Rf = 0.6 (silica gel, 15:9:1:2 v/v/v/v CHCl$_3$/MeOH/AcOH/ H$_2$O); Retention time (Gradient 1): 8.98 min; HRMS (+ESI) $m/z$ for $C_{14}H_{25}N_4O_7$ [M + H]$^+$ calculated 361.1718, observed: 361.1709.

### N-acetylglucosaminyl-β(1-4)-N-acetylmuramoyl-l-alanyl-γ-d-glutamyl-l-alanyl-d-alanine (1)

Disaccharide **6** (149 mg, 0.3 mmol) was dissolved in DMF (1 mL) with stirring under argon. N-Methylmorpholine (33 μL, 0.3 mmol) followed by bis(pentafluorophenyl) carbonate (118 mg, 0.3 mmol) were successively added to the solution of disaccharide and the reaction mixture was stirred at ambient temperature under argon. The reaction was complete in 1 h as indicated by the disappearance of **6** (TLC: silica gel, 2.5:2.2:0.2:0.25 v/v/v/v CHCl$_3$/MeOH/AcOH/H$_2$O, staining with phosphomolybdic acid; for **6**: Rf = 0.5, for (presumable) pentafluorophenyl ester of **6**: Rf = 0.65). Pyridine (30 μL, 0.375 mmol) followed by tetrapeptide **5** (119 mg, 0.25 mmol) were added to the reaction mixture with stirring. After the complete dissolution of **5** stirring was turned off and the reaction was allowed to proceed for 16 h. The reaction mixture was added dropwise to ethyl acetate (40 mL) and the suspension was centrifuged. The solid was resuspended and centrifuged in EtOAc (3 × 10 mL), MeCN (3 × 10 mL), ether (3 × 10 mL). The remaining solid was dried in vacuo to provide 210 mg of crude product. Compound **1** (46 mg, 22%) was isolated using RP-HPLC (2.1 × 25 cm C18 column, isocratic elution with 6% MeCN in water (0.1% TFA)). For compound **1**: Retention time (Gradient 2): 9.47 min (isomer 1), 9.77 min (isomer 2); HRMS (+ESI) $m/z$ for $C_{33}H_{54}N_6O_{19}$ [M + H]$^+$ calculated 839.3517, observed: 839.3488.

### Recombinant l-alanoyl-d-glutamate peptidases of colibacteriophages T5, RB43, and RB49

Active homogeneous preparations of the recombinant enzymes EndoT5, EndoRB43 and EndoRB49 were obtained from producer strains obtained earlier,

using the previously developed method (*Mikoulinskaia et al., 2018*) and stored frozen at −20 °C.

## NMR-based analysis of the GMTP degradation

NMR measurements were conducted on the AVANCE 600 III spectrometer (Bruker), with an operating frequency of 600 MHz, at 298 K, a spectral width of 24 ppm and the 90-degree pulse of 11 μs, number of scans 32–128, depending on the value of the signals required for the processing. NOE spectra in the spin diffusion mode were excited at a frequency of the methyl pool of the protein ~0.8 ppm. To excite the effect, we used power 65–70 db, relaxation delay 3 s, excitation time 0.5 s, mixing time 0.08 s, number of accumulations 1000–4000. To assign lines in the spectrum of the glycopeptide, two-dimensional experiments were carried out: $^1$H-$^{13}$C-HSQC, $^1$H-$^{13}$C-HMBC, $^1$H-$^1$H-COSY, $^1$H-$^1$H-TOCSY with standard pulse sequences of the Bruker library.

GMTP was dissolved in Tris-HCl buffer (50 mM, pH 7.3) or sodium acetate buffer (50 mM, pH 4.3) at a concentration of 2.2 mg/mL (2.63 mM) and a spectrum was recorded, then the enzymes were added, the final concentrations of which were in the 0.19–0.21 mg/mL range (0.0128–0.0142 mM, such enzyme concentrations are sufficient for their observation in NMR spectra). Then, the spectra were recorded until the completion of the hydrolysis process. The temperature at recording of the NMR spectra was 298 K.

## RESULTS AND DISCUSSION

The preparation of target compound **1** involved chemical synthesis of protected tetrapeptide **4** from two peptide fragments **2** (*Denoel et al., 2014*) and **3** (*Reid et al., 2019*) followed by global deprotection and selective acylation of the amino group in the intermediate **5** with pentafluorophenyl ester of N-acetylglucosaminyl-(β1-4)-N-acetylmuramic acid. The activated ester was prepared from the free acid **6** using bis (pentafluorophenyl) carbonate following the procedure described for esterification of the analogous saccharide (*Jezek et al., 1990*). Scheme of this synthesis is shown in Fig. 1.

In this way, we have synthesized a glycopeptide that is N-acetylglucosaminyl-β(1-4)-N-acetylmuramyl-l-alanyl-γ-d-glutamyl-l-alanyl-d-alanine (glucosaminyl muramyl tetrapeptide, GMTP) with a molecular weight of 840 Da. GMTP is a water soluble compound that simulates the glycopeptide region of the peptidoglycan of the bacterial cell wall, containing the l-Ala-d-Glu dipeptide, which is hydrolyzed by the bacteriophage peptidases EndoT5, EndoRB43, and EndoRB49. The structure of GMTP is shown in Fig. 2, whereas Table 1 represents the assignments of the NMR lines, which we obtained based on the analysis of the NMR spectra: $^1$H -$^{13}$C-HSQC, $^1$H-$^{13}$C-HMBC, $^1$H-$^1$H-COSY, $^1$H-$^1$H-TOCSY.

When a homogeneous recombinant enzyme preparation is added to the GMTP solution, the degradation of a substrate begins by the hydrolysis of the peptide bond between l-Ala$_1$ and d-Glu$_2$. The cleavage of this bond is evidenced by the characteristic changes in the NMR spectra, such as: (1) the presence of the C-terminal carboxyl belonging to the residue l-Ala$_1$; (2) the absence of the signal of the amide proton of d-Glu$_2$;

**Figure 1 Scheme representing synthesis of the endolysin substrate mimetic 1.** (A) CF3CO2Pfp, Py, DCM; (B) **3**, NMM, MeCN, 62 %; (C) TFA, H2O, 92 %; (D) (PfpO)$_2$CO, NMM, DMF; (E) **5**, Py.

**Figure 2 Chemical structure if the N-acetylglucosaminyl-β(1-4)-N-acetylmuramyl-l-alanyl-γ-d-glutamyl-l-alanyl-d-alanine (glucosaminyl muramyl tetrapeptide, GMTP).** The numbers correspond to the sequential numbers of the amino acid residues. The peptide bond subjected to hydrolysis is colored in green.

(3) disappearance of the unique spatial environment of the $\beta_{1,2}$-protons of the d-Glu$_2$ residue; (4) significant changes in the positions of the remaining signals of the l-Ala$_1$ and d-Glu$_2$ residues. In addition, there is a change in the chemical shift of the signal of the α-anomeric proton of α-N-acetylmuramic acid, as shown in Fig. 3, as well as the changes in the integral intensity, which make it possible to follow the kinetics of the cleavage process, as shown in Figs. 4A–4D.

Figure 4A represents kinetic curves showing a decrease in the concentration of the initial GMTP (descending), when it is cleaved by EndoRB49 peptidase at the peptide bond l-Ala-d-Glu and its transformation into two products: N-acetylglucosaminyl-β(1-4)-N-acetylmuramoyl-l-alanine and γ-d-glutamyl-l-alanyl-d-alanine, the increase in the concentration of which is reflected by the rising curve. The measurements were carried out using the signals of the anomeric sugar proton at ~5.4 ppm, which are shifted relative to

**Table 1 Signal assignment for N-acetylglucosamine-β(1-4)-N-acetylmuramyltetrapeptide.**

| Spin system | Atom name | Chemical shift $^1$H (ppm) | Chemical shift $^{13}$C (ppm) |
|---|---|---|---|
| β-N-acetylglucosamine | 1′H; 1′C | 4.544 | 102.987 |
| | 2′H; 2′C | 3.750 | 58.803 |
| | 3′H; 3′C | 3.431 | 73.035 |
| | 4′H; 4′C | 3.567 | 76.352 |
| | 5′H; 5′C | 3.434 | 78.804 |
| | 6′H; 6′C | 3.775 | 63.923 |
| | HN$_{Ac}$ | 8.410 | |
| | ′C (C = O)$_{Ac}$ | | 177.519 |
| | H; C (CH$_3$)$_{Ac}$ | 2.055 | 24.888 |
| α-N-acetylmuramic acid | 1′H; 1′C | 5.222 | 92.903 |
| | 2′H; 2′C | 3.832 | 56.352 |
| | 3′H; 3′C | 3.878 | 73.683 |
| | 4′H; 4′C | 3.770 | 79.064 |
| | 5′H; 5′C | 3.873 | 78.102 |
| | 6′H; 6′C | 3.826 | 62.548 |
| | HN$_{Ac}$ | 7.906 | |
| | ′C (C = O)$_{Ac}$ | | 176.847 |
| | H; C (CH$_3$)$_{Ac}$ | 1.950 | 24.842 |
| | H; C (CH)$^a_{Lac}$ | 4.544 | 80.359 |
| | H; C (CH$_3$)$^a_{Lac}$ | 1.404 | 20.769 |
| | ′C (C = O)$^a_{Lac}$ | | 177.993 |
| β-N-acetylmuramic acid | 1′H; 1′C | 4.648 | 97.778 |
| | 2′H; 2′C | 3.720 | 58.399 |
| | 3′H; 3′C | 3.857 | 78.105 |
| | 4′H; 4′C | 3.599 | 81.948 |
| | 5′H; 5′C | 3.482 | 77.834 |
| | 6′H; 6′C | 3.886 | 62.604 |
| | HN$_{Ac}$ | 7.706 | |
| | ′C (C = O)$_{Ac}$ | | 176.958 |
| | H; C (CH$_3$)$_{Ac}$ | 1.977 | 24.842 |
| | H; C (CH)$^a_{Lac}$ | 4.544 | 80.359 |
| | H; C (CH$_3$)$^a_{Lac}$ | 1.404 | 20.769 |
| | ′C (C = O)$^a_{Lac}$ | | 177.993 |
| l-Ala$_1$ | HN | 8.433 | |
| | αH; αC | 4.329 | 52.685 |
| | βH; βC | 1.442 | 19.737 |
| | ′C (C = O) | | 176.735 |
| d-Glu$_2$ | HN | 8.027 | |
| | αH; αC | 4.276 | 57.120 |
| | βH$_{1,2}$; βC | 1.935; 2.125 | 30.912 |
| | γH$_{1,2}$; γC | 2.326 | 34.658 |
| | ′C (C = O)$_{B-b}$ | | 176.775 |
| | ′C (C = O)$_{S-ch}$ | | 178.503 |

| Spin system | Atom name | Chemical shift $^1$H (ppm) | Chemical shift $^{13}$C (ppm) |
|---|---|---|---|
| l-Ala$_3$ | HN | 8.238 | |
| | αH; αC | 4.279 | 52.702 |
| | βH; βC | 1.378 | 19.579 |
| | 'C (C = O) | | 176.776 |
| d-Ala$_4$ | HN | 7.878 | |
| | αH; αC | 4.156 | 53.648 |
| | βH; βC | 1.315 | 20.270 |
| | 'C (C = O) | | 182.413 |

**Notes:**
[a] The chemical shifts for lactate nuclei are indistinguishable for α-and β-isomers of N-acetylmuramic acid;
[b] Green font is used to indicate the positions of the signals from the hydrolyzed bond highlighted in green in Fig. 2.

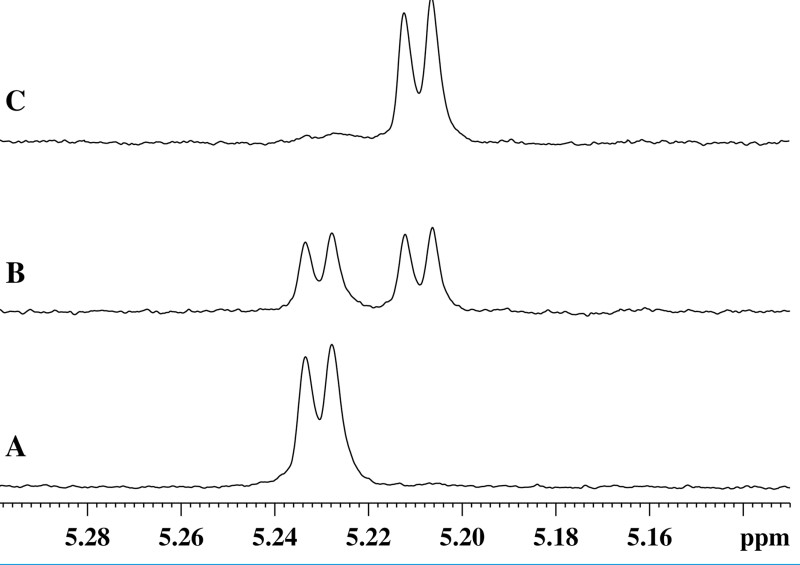

**Figure 3 Stages of changes in the $^1$H-NMR spectrum of the region of absorption of the α-anomeric proton of α-N-acetylmuramic acid of GMTP depending on time after the addition of enzymes.** (A) Spectrum taken immediately after the addition of any of the studied peptidases. (B) $^1$H-NMR spectrum corresponding to the time sufficient for the conversion of 50% of the initial GMPT into N-acetylglucosamine-N-acetylmuramyl-l-alanine. (C) Spectrum corresponds to the time of almost complete transformation of the initial GMPT to N-acetylglucosamine-N-acetylmuramyl-l-alanine.

each other and clearly distinguishable for these two products. The fitting of the experimental data was carried out by simple exponents: $f(t) = a \times e^{-\lambda \times t}$ and $f(t) = a \times (1 - e^{-\lambda \times t})$ for the ascending and descending curves, respectively (here, $\lambda$ is the decay constant).

Therefore, to characterize the substrate decay rate, we used the accepted standard parameter, half-life time, defined as $t_{1/2} = ln2/\lambda$. In the case of the EndoRB49-catalyzed GMTP hydrolysis, the $t_{1/2}$ value was 36 min. Figure 4B shows the results of similar experiments for EndoT5 peptidase; i.e., where the GMTP concentration was decreasing

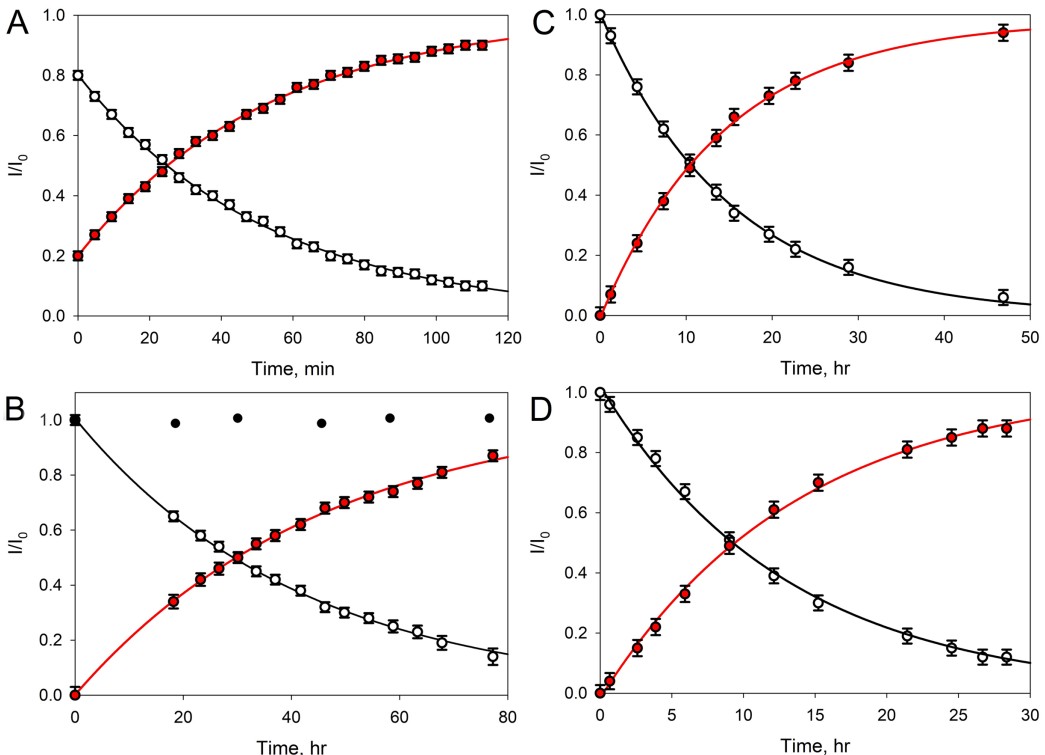

**Figure 4 Transformation of GMTP with the addition of enzymes.** The ascending curve shows an increase in the concentration of the final product, and the descending curve shows the change in the concentration of the original substrate. The circles are the measured values, the lines are the exponential approximations (fitting) ($R^2$ = 0.9995). In these experiments, the GMTP hydrolysis was catalyzed by (A) EndoRB49 (pH = 7.3); (B) EndoT5 (pH = 7.3). Black circles show to the lack of activity of an inactivated mutant EndoT5 with the point substitution of the catalytic aspartate (EndoT5D130A); (C) EndoRB49 (pH 4.3); and (D) EndoRB43 (pH = 7.3).

because of its cleavage by the EndoT5 peptidase. For EndoT5 under the same conditions $1/\lambda$ = 49.19 h and $t_{1/2}$ = 34.10 h. It should be noted that the addition of a mutant EndoT5 with a point substitution of the catalytic aspartate (EndoT5D130A) to the GMTP solution did not cause substrate hydrolysis, in contrast to the wild type protein (see black circles in Fig. 4B). Figure 4C shows the curves of the GMTP hydrolysis by EndoRB49 peptidase at pH 4.3, which is outside the physiological pH optimum of this enzyme. In fact, the pH optimum for all three studied enzymes is wide and lies in the range of 7.0–9.0 (*Mikoulinskaia et al., 2018*; *Mikoulinskaia et al., 2013*). Therefore, pH 7.3 and pH 8.0 that were used in previous work for the analysis of catalytic reactions of EndoT5, EndoRB43, and EndoRB49 (*Shadrin et al., 2020*) are within this pH optimum range, whilst pH 4.3 is the value at which enzymes are practically devoid of activity. It is these pH values that were used for structural NMR experiments, since they provide the minimum proton exchange of peptide bonds and the maximum protein solubility, which significantly improves the quality of NMR spectra and reduces the time required to obtain a complete set of spectra for structural analysis. The very high rate of hydrolysis of GMTP by the EndoRB49 enzyme suggested that it might have residual catalytic activity even at pH 4.3, which was in fact demonstrated. This eliminates the question of the adequacy of the

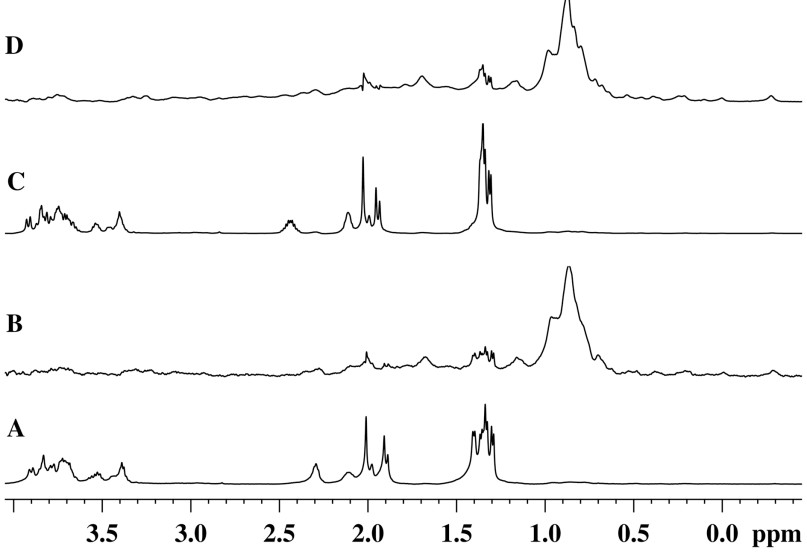

**Figure 5** <sup>1</sup>H-NMR and <sup>1</sup>H-NOE spectra of the aliphatic region of the EndoRB49 enzyme upon interaction with GMTP in D$_2$O, pH 4.2, T = 295K. (A) $^1$H-NMR (beginning of transformation); (B) $^1$H-NOE at saturation of the signal of the methyl pool of the enzyme (~0.8 ppm) (beginning of transformation); (C) $^1$H-NMR (end of transformation); (D) $^1$H-NOE at saturation of the signal of the methyl pool of the enzyme (~0.8 ppm) (end of transformation).

conditions for obtaining the structural, dynamic and catalytic characteristics, which is very important for future structural studies of the protein EndoRB49, which is extremely interesting and promising in terms of biotechnological applications. It is also important to emphasize that the residual activities of other investigated orthologous peptidases-EndoT5 and EndoRB43-at pH 4.3 are too low to be detected. Figure 4C illustrates the fact that with a decrease in pH from the values optimal for the enzyme, the rate of hydrolysis sharply decreases, and in this case $t_{1/2}$ = 14.77 h. Finally, Fig. 4D shows the curves reflecting the change in the initial GMTP concentration upon its cleavage by the EndoRB43 peptidase. At a EndoRB43 concentration of 0.19 mg/mL, $t_{1/2}$ = 8.8 h.

A four-fold increase in the protein concentration at a constant substrate concentration naturally led to a decrease in the $t_{1/2}$ value. In our study, the half-life decreased from $t_{1/2}$ = 8.8 h to $t_{1/2}$ = 2.1 h for EndoRB43. Since the protein retains its properties under the same environmental conditions, the coefficient $\lambda$ must remain the same, which means that the coefficient $a$ must change and depend on the concentrations of the protein and substrate. Based on our data, this means that the coefficient $a$ = C$_{substrate}$/C$_{protein}$ almost linearly depends on the concentrations: an increase in the protein concentration leads to a decrease in $a$, and hence the half-life time ($t_{1/2}$), an increase in the substrate concentration leads to an increase in $a$ and to an increase in $t_{1/2}$. In our study, a 4-fold decrease in the protein concentration (C$_{substrate}$/C$_{protein}$/4 = 4 × C$_{protein}$/C$_{substrate}$) = 4$a$, led to a 4-fold increase in the GMTP half-life.

Figures 5A and 5C show the $^1$H-NMR spectra of the GMTP glycopeptide in the presence of the zinc-containing form of the most active enzyme, EndoRB49. The spectra of

the aliphatic region show mostly GMTP signals, since the protein signals are much lower in amplitude and therefore practically are not detectable. In the $^1$H-NOE spectra upon excitation at a frequency of the protein methyl pool of ~0.8 ppm (Figs. 5B and 5D), protein signals appear due to spin diffusion (*Melnik et al., 2011*). In addition, one can clearly see prominent responses of the GMTP alanine signals in the range of 1.45–1.25 ppm. These signals-responses appear when the excitation energy is transferred from the protein molecule when it reaches the GMTP molecule, as a result of which the hydrolysis of the l-Ala-d-Glu peptide bond occurs (Fig. 5C). As the peptide transforms, these signals decrease in intensity and change their position corresponding to alanine signals in the resulting peptide without the sugar part. With the complete cleavage of GMTP molecules by the enzyme, the spectra have the form shown in Figs. 5B and 5D.

Therefore, the alanine part of GMTP serves as an "anchor site" defining orientation of the glycopeptide and providing its retention sufficient for the hydrolysis that is captured by a protein molecule for the rapprochement with the substrate mimetic and subsequent enzymatic reaction. The signal at 1.4 ppm belongs to the methyl group of N-acetylmuramic acid, and the signal at 1.44 ppm belongs to the methyl group of $Ala_1$ (see Table 1). This means that just in this region the distance between the enzyme molecule and the glycopeptide is the smallest, which most likely indicates the position of the interaction site near the N-acetylmuramic acid of the glycopeptide. However, the excitation energy transfer is not observed for the signals from the carbohydrate residues of the glycopeptide. Therefore, the interaction in this region is almost pointwise. This site of interaction on the glycopeptide is common to all of the enzymes analyzed in this study and does not depend on the rate of the transformation reaction. The protein molecule retains the affinity for the methyl part of all alanine residues after the cleavage of GMTP, since the energy transfer from the protein occurs even after the end of the enzymatic reaction, as can be seen in the $^1$H-NOE spectrum in Fig. 5D. However, it is possible that this is no longer a specific interaction, but a simple hydrophobic interaction of alanines with a protein molecule.

It is difficult to say to what extent this interaction takes place in a natural substrate, the structure of which differs significantly from an artificial glycopeptide. For example, in position 3 of the peptide subunit, l-alanine is practically absent in bacteria, and in peptidoglycan of the A1γ type, characteristic of the host of bacteriophages T5, RB43 and RB49-*Escherichia coli*-in position 3 is meso-diaminopimelic acid [2]. It is very likely that this has a strong effect on the rate of hydrolysis of GMTP and on the process of substrate binding, taking into account the hydrophobic interaction of the reaction products with the enzyme molecule after the hydrolysis and the hindered dissociation of the enzyme-product complex. Nevertheless, as it was shown, the glycopeptide developed in this study-N-acetylglucosaminyl-β(1-4)-N-acetylmuramyl-l-alanyl-γ-d-glutamyl-l-alanyl-d-alanine-may well be used as one of the mimetic peptidoglycans for modeling its hydrolysis in vitro.

The highest activity of EndoRB49 towards GMTP is in good agreement with the data indicating the low selectivity of this enzyme in relation to the cell walls of bacteria with variations in the structure of peptidoglycan. Thus, it was recently shown (*Shadrin et al.,*

*2020*) that this enzyme, with the highest rate among the studied enzymes, hydrolyzes the cell wall of not only Gram-negative bacteria, but also Gram-positive bacteria of the genera *Bacillus*, *Cellulomonas* and *Sporosarcina*, whose peptidoglycans had different structures (A1γ, A4α, A4β) and chemical modifications (amidation). However, it is clear that to demonstrate conclusively that the enzymes differ in their selectivity toward peptidoglycan would require additional experiments performed using purified peptidoglycan or specific muropeptides (which is outside the scope of the current study).

It seems that our data suggest that the methyl groups of all three alanine residues can act as anchor points for protein binding. This raises an important question on the biological relevance of binding to the methyl group of the l-Ala at position three of the peptide stem, as this amino acid is not present in naturally occurring muropeptides. To answer this question one should remember that in naturally occurring muropeptides, a rather extended hydrophobic side chain region is always present in position 3. This was modeled here using a pair of alanines, which made it possible to show the process of catalysis and substrate binding by the proteins under study. It should also be noted that any model that is closest to nature is nothing more than a simplification justified by certain considerations. To study the various molecular details of the catalytic process, such as substrate binding, enzyme-substrate affinity, reaction rate, processivity and selectivity of the enzyme, mimetics of various structures can be optimal, and only a comparative analysis will help uniting the results of experiments and specific problems.

It would be interesting and important to know whether the disaccharide portion of the mimetic is required for activity. However, there is evidence that it is so. First, while the NOE spectra do not show signals from the protons of the sugar rings, which means that they are quite far from the protein globule, the protons of the methyl groups indicated in Fig. 2 as "NHA" contribute to the spectrum in the range 2–1.9 ppm (Fig. 5B), although these signals disappear at the end of the hydrolysis process. This suggests that the whole glycopeptide is oriented relative to the protein molecule in such a way that its hydrophobic groups come close to the protein surface, fixing positioning of the glycopeptide relative to protein globule by hydrophobic interactions so that the disaccharide forms a rather large angle with the protein surface. Moreover, the amino group of alanine, that would appear in the absence of an amide bond with muramic acid, is likely to inhibit with hydrolysis in the same way as did amidation of the C-terminal glutamate in GMDP (this biologically active mimetic, as we have verified, is not hydrolyzed by the studied enzymes, despite the similarity with natural PG). However, the assumption about the importance of the disaccharide part requires additional experimental support, and for the relevance of such an experiment, we are planning to use not only molecule 5, but also other relevant molecules such as shortened minimal substrates-four-membered species with different structures (disaccharide dipeptides).

## CONCLUSIONS

In this work, one of the simplest models mimicking the peptidoglycan fragment of the bacterial cell wall, glycopeptide N-acetylglucosaminyl-β(1-4)-N-acetylmuramyl-l-alanyl-γ-d-glutamyl-l-alanyl-d-alanine, was successfully synthesized. This water soluble

peptidoglycan mimetic and its in vitro degradation by the lytic l-alanoyl-d-glutamate peptidases of colibacteriophages T5, RB43, and RB49 (EndoT5, EndoRB43, and EndoRB49, respectively) were characterized by the solution NMR. It is shown that this model peptidoglycan can be hydrolized by EndoT5, EndoRB43, and EndoRB49, with the cleavage efficiency being enzyme- and environmental pH-dependent. It was also shown that to hydrolyze this glycopeptide mimetic, enzymes approach the glycopeptide near the methyl groups of all three alanines. Therefore, developed in this study glycopeptide may serve as a suitable model for the analysis of the peptidoglycan hydrolysis in vitro.

### Funding
This work was supported by a grant from the Russian Foundation for Basic Research (RFFI N 18-04-00-492; Galina Mikoulinskaia). There was no additional external funding received for this study. The funders had no role in study design, data collection and analysis, decision to publish, or preparation of the manuscript.

### Grant Disclosures
The following grant information was disclosed by the authors:
Russian Foundation for Basic Research: RFFI N 18-04-00-492.

### Competing Interests
Vladimir N. Uversky is an Academic Editor for PeerJ.

### Author Contributions
- Viatcheslav Azev performed the experiments, analyzed the data, prepared figures and/or tables, authored or reviewed drafts of the paper, and approved the final draft.
- Alexey Chulin performed the experiments, analyzed the data, prepared figures and/or tables, authored or reviewed drafts of the paper, and approved the final draft.
- Maxim Molchanov performed the experiments, analyzed the data, prepared figures and/or tables, authored or reviewed drafts of the paper, and approved the final draft.
- Dmitry Prokhorov conceived and designed the experiments, performed the experiments, analyzed the data, prepared figures and/or tables, authored or reviewed drafts of the paper, and approved the final draft.
- Galina Mikoulinskaia conceived and designed the experiments, performed the experiments, analyzed the data, prepared figures and/or tables, authored or reviewed drafts of the paper, and approved the final draft.
- Vladimir N. Uversky conceived and designed the experiments, authored or reviewed drafts of the paper, and approved the final draft.
- Viktor Kutyshenko conceived and designed the experiments, analyzed the data, prepared figures and/or tables, authored or reviewed drafts of the paper, and approved the final draft.

## Data Availability

NMR spectra and HPLC chromatograms of the synthesized compound, and raw measurements, are available in the Supplemental Files.

## Supplemental Information

Supplemental information for this article can be found online at http://dx.doi.org/10.7717/peerj.11480#supplemental-information.

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
