# Peer review of "Chemical synthesis of peptidoglycan mimetic–disaccharide-tetrapeptide conjugate and its hydrolysis by bacteriophage T5, RB43 and RB49 L-alanyl-D-glutamate peptidases"

_PeerJ, doi:10.7717/peerj.11480_

## Round 0.1 · original submission · Major Revisions

The two reviewers gave thorough reviews and identified several key concerns. I urge you to pay close attention to these comments in your revised manuscript.

Reviewer 1 ·

Basic reporting

V. Azev et al describes the synthesis and the hydrolysis by a set of peptidases of one disaccharide-tetrapeptide. This disaccharide peptide is presented as peptidoglycan mimetic which could be used for hydrolysis study by spectroscopy methods. The English language is clear and unambiguous. The introduction needs to be more documented, especially concerning the structure of peptidoglycan to correctly introduce the choice of the synthesized compound. A review could be found easily on Pubmed concerning PG structures.

Authors claim that “the process of the peptidoglycan binding and hydrolysis by enzymes cannot be studied by the spectral methods”. This assertion must be nuanced [For spectral methods with PG or PG fragments see Bougault et al, J. Struct. Biol. (2019) 206(1):66-72; Forrest et al, J. Biol. Chem. (1991) 266(36):24485-91 or Maya-Martinez et al, Front. Microbiol. (2018), 9:3223].

The authors should explain more clearly the advantage of their methodology. For example, the incorporation of the disaccharide moiety described in the paper seems the most important issue for the obtention of mg of PG analogs.

Figures are relevant even if Fig. 1 and Fig. 3 appear blurry.

Figure 3 must be reorganized: panel C representing the beginning of the reaction must be on the upper part of the figure and renamed panel A. Actual panel A, representing the end of the reaction, must be on the lower part of the figure and must be renamed panel C.

The same modifications must be done for figure 5.

Experimental design

The experiments are well described. Even if the chemistry part is classical, purity analysis performed by TLC for compound 4 seems insufficient for publication. Nor rp-HPLC chromatograms or HRMS spectra are reported for the final product (in the main text or in the supplementary Data). This point is essential to estimate purity products and need improvement for publication.

Validity of the findings

- Hydrolysis of the amide bond between L-Ala1 and D-iGlu2 is followed by NMR, it seems that an identification of the products could be easily done by MS. These experiments would strengthen the results presented in this article.

- Concerning the disaccharide moiety, it would be interesting to know if this part of the molecule is necessary for the enzymatic activity. Molecule 5 could be used for such a purpose. This study could be introduced in the Discussion section (L253-255)

- Concentrations of GMTP and protein should be given in molar in the NMR-based analysis of the GMTP degradation to indicate the turnover of the reaction during experiments for proper comparison between different proteins used in this study (concentrations are given only in mg/mL and no indication is given concerning the molecular weight of the proteins).

Reviewer 2 ·

Basic reporting

1) Background and context: The authors justify the synthesis of the soluble GMTP muropeptide mimetic as follows: “because of the insolubility of peptidoglycans…” they “cannot be studied by spectral methods” (Line 34 and 35). In addition, “It is impossible to study the process of the substrate binding and hydrolysis by enzymes by the spectral methods… since natural peptidoglycans are insoluble” (Line 64 and 65).

These statements are true. However, the authors did not mention that soluble muropeptides can be generated from insoluble peptidoglycan by digestion with mutanolysin or cellosyl, then purified by RP-HPLC (used previously as a source of soluble muropeptides for NMR experiments, e.g. Atrih, et al. Journal of bacteriology, 1999. doi:10.1128/JB.181.13.3956-3966.1999). There are also commercially available soluble muropeptides such as MDP, Mtri-DAP and Mtri-Lysine (such as those produced by Invivogen), which include dibasic amino acids (mesoDAP and Lysine) at position 3 of the peptide stem that are found in naturally occurring peptidoglycan, but absent from the GMTP structure generated in this paper.

The authors should include additional justification for the study. What is the advantage of using GMTP, which has an unusual L-Alanine at position 3, relative to alternatives such as MDP, Mtri-Dap or Mtri-Lysine?


2) Article structure: The article is generally well structured. Regarding the experiment in which GMTP digestion by the 3 enzymes is compared (lines 206-237), the order in which the data are presented is confusing. EndoRB49 activity is first compared to EndoT5 at pH 7.3 (Panels A and B), then EndoRB49 activity compared at pH 7.3 and pH 4.3 (panel C). Finally, EndoRB43 activity at pH 7.3 is presented (Panel D).

It would be more logical for the authors to present and discuss all of the experiments performed at pH 7.3 together. The activity of EndoRB49 at pH 7.3 and pH 4.3 should be discussed separately. The order of the panels in figure 4 should be altered accordingly. It would aid reading if the enzyme and pH were annotated directly on the relevant panel.


3) “Therefore, the alanine part of GMTP is an “anchor site”…” (Line 250).
This statement is ambiguous. It is unclear whether the authors are referring to all 3 alanine residues, or a specific alanine – if so which one? The authors should clarify this point.


4) Regarding lines 46-47: “…suggesting that EndoRB49 is characterized by the lowest selectivity, and hence the potentially broader spectrum of the peptidoglycan types subjected to hydrolysis.” One of the authors previously published on this topic, in which lysis of bacteria with different peptidoglycan types by the three enzymes was assessed. This citations should be added here. (Shadrin et al. J Appl Microbiol. 2020. doi: 10.1111/jam.14910).


5) Clarity of writing: There are many spelling errors and grammatical issues throughout, particularly in the methods and materials. The paper should undergo a substantial redraft to reach publication quality. There are too many errors to provide a complete list, but examples include:

Line 35: “…studied by the spectral methods” should be “…studied by spectral methods”.
Line 61: “…the L-Ala-D-Glu bond is always exist. It is this bond is the target…” should be “… the L-Ala-D-Glu bond is always present. This bond is the target…”
Line 86: Should read “…to a residual solvent signal.” or “…to residual solvent signals.”
Line 87 & 88: “standart” should be “standard”
Line 105: “evoparated” should be “evaporated”
Line 103 & 105 “resulted” should be “resulting”
Line 108: “homogenious” should be “homogeneous”
Line 193: The structure is referred to as “GSCP” instead of “GMTP”
Line 115: “(3 2 mL)” should be “(32 mL)” or “(3.2 mL)”
Line 177: “…until the complete completion…” should be “…until completion…”


6) Labelling of Figure 3. The labelling of the spectra is strange, as is the order they are presented in the figure legend. Why does it begin at C and end on A? The order should be changed to A, B, C.

Experimental design

The experiments appear to have been conducted to a high standard, and described in sufficient detail to be replicated.

My concern in terms of experimental design relates to the issue raised in "basic reporting". A justification should be given for using GMTP as an alternative to soluble muropeptides, that contain the dibasic amino acids found in natural peptidoglycan.

Validity of the findings

1) Regarding lines 46-47, and the work published by Shadrin et al. 2020. (doi: 10.1111/jam.14910). I do not agree the faster activity of EndoRB49 towards GMTP shown in the current study “suggests” it is “characterized by the lowest selectivity”. It would be more correct to say that it is “consistent” with this hypothesis, which was put forward in the paper of Shadrin et al. The authors should rephrase this sentence accordingly.

Care should be taken, as (unless I'm mistaken) the paper of Shadrin et al. studied lysis of bacteria by the three L-D peptidases, and therefore the presence of other wall polymers such as teichoic acids or capsule could have affected the enzyme activity. To demonstrate conclusively that the enzymes differ in their selectivity toward peptidoglycan, would require additional experiments performed using purified peptidoglycan or specific muropeptides (which is outside the scope of the current manuscript).


2) In this manuscript, and the paper of Shardin et al, the reactions were performed at pH 7.3 and pH 8 respectively. In this manuscript, EndoRB49 was also tested at pH 4.3, but it seems that EndoT5 and EndoRB43 were only tested at pH 7.3. What is the pH optimum for EndoT5 and EndoRB43? Is a pH of 7.3 close enough to the pH optimum of all three enzymes that it would not to be responsible for the difference in reaction rates? It would be useful to address this point in the manuscript.


3) If I have understood correctly, the authors say that the methyl groups of all 3 alanine residues act as anchor points for protein binding. In this case, what is the biological relevance of binding to the methyl group of the L-Alanine at position 3 of the peptide stem, as this amino acid is not present in naturally occurring muropeptides? The authors should comment on this.


4) Lines 219 to 222. A good negative control is mentioned here (catalytically inactived T5 enzyme) but the data is not shown. The authors should add it to figure 4, or included it as supplementary data.

---

## Round 0.2 · Minor Revisions

The reviewer suggests that you correct minor issues with Fig 3 and some typographic errors. Once these issues are addressed, the manuscript could be ready for publication.

Reviewer 1 ·

Basic reporting

The requests for corrections to the manuscript by Azev et al have been made.

- The introduction was modified as requested to introduce in a more documented way the molecule synthesized in this study.
- The authors have also discussed the interest of introducing an L-Alanyl moiety at the third position of the synthesized peptide.
- The authors added a discussion part in the body of the text concerning the importance of the disaccharide moiety of their compound.

- All corrections concerning figures are correctly introduced in the text.

Experimental design

- Some missing characteristics (mass and rpHPLC data) were added in SI as requested.
- The authors claim that some technical limitation of the LC-MS system they used does not allow identification of the reaction products. They explained that the reaction products are not retained on a C18 column and could not be analyzed by mass spectroscopy. This observation is effectively possible (personal observation on similar molecules). Direct injection on MS could be done after precipitation of proteins but NMR data are sufficiently convincing for publication.

Validity of the findings

no comment

Additional comments

In conclusion, I think this work is suitable for publication.

Reviewer 2 ·

Basic reporting

The authors have addressed all the comments satisfactorily. There is only one issue regarding figure 3, and some minor orthographic/english errors throughout the manuscript which should be corrected before publication. I have tried to provide a complete list below.

Otherwise the manuscript has been completed to a high standard.
* * *
In the uploaded figure files, and review PDF, the figure 3 legend is incorrect. It has been duplicated from figure 2. The correct legend is found in the Word document however. Please ensure the correct legend is included in the published manuscript, particularly as there was a correction made regarding the order in which the data was presented.

The incorrect legend: “Chemical structure if the N-acetylglucosaminyl-β(1-4)-N-acetylmuramyl-l-alanyl-γ-d-glutamyl-l-alanyl-d-alanine (glucosaminyl muramyl tetrapeptide, GMTP).
The numbers correspond to the sequential numbers of the amino acid residues. The peptide bond subjected to hydrolysis is colored in green.”



63: “linked by the interpeptide bridges” should be “linked by interpeptide bridges”

69: “(for example, 1 corresponds to the bridge that does not contain amino acids)” – a little strange to say a bridge with no amino acids. Better so say “(for example, 1 corresponds to direct cross-linking between peptides stems, with no bridging
amino acids).”

71: “…acid, DAP). . The amino acid…” should be “…acid, DAP). The amino acid…”

75: “…the L-Ala-D-Glu bond is always exist.” - “… the L-Ala-D-Glu bond is always present.”

96: Double space.

105: “…specific activity on natural substrate.” Should be “…specific activity on the natural substrate.”

120: “…by the spectral methods” shoud be “by spectral methods”.

214 and 231: Values are given as (3 1.5 ml) (3 10 mL) (3 10 mL) (3 10 mL). What does the 3 signify? Should it read 3x?

256 and 258: since the units are mM, the “/L” is unnecessary. Alternatively use the units “mmol/L”.

296: “acetylmuramil” should be “acetylemuramyl”

307: “…noted that the addition of a mutant EndoT5 with the point substitution of the catalytic aspartate (EndoT5D130A) to the GMTP solution did not cause the substrate hydrolysis” should be “…noted that the addition of a mutant EndoT5 with a point substitution of the catalytic aspartate (EndoT5D130A) to the GMTP solution did not cause substrate hydrolysis”

312: “Therefore, although pH 7.3 and pH 8.0 that were used in previous work for the analysis of catalytic reactions of EndoT5, EndoRB43, and EndoRB49 (Shadrin et al. 2020) are within this pH optimum range, and pH 4.3 is the value at which enzymes are practically devoid of activity.” Should be “Therefore, pH 7.3 and pH 8.0, that were used in previous work for the analysis of catalytic reactions of EndoT5, EndoRB43 and EndoRB49 (Shadrin et al. 2020), are within this pH optimum range, whilst pH 4.3 is the value at which the enzymes are practically devoid of activity.”

325: “…which is very important for future structural studies of the extremely interesting and promising for biotechnology protein EndoRB49.” Should be “…which is very important for future structural studies of the protein EndoRB49, which is extremely interesting and promising in terms of biotechnological applications.”

380: “Nevertheless, as it was shown, the developed in this study glycopeptide…” should be “Nevertheless, as it was shown, the glycopeptide developed in this study…”

407: “However, there is evidence that it is so.” Remove the extra spaces.

566: Figure 4 legend, “Black circles show…” should be “Black, filled circles show…” since there are two sets of black circles representing different data.

568: “C. EndoRB49 (pH 4.3); C. EndoRB49 (pH 4.3);” text duplication.

Experimental design

The experimental design and reporting is to a high standard

Validity of the findings

The underlying data is provided in a robust manner. Conclusions and speculative discussion are clear.

---

## Round 0.3 · accepted · Accept

We look forward to working with you on your future contributions.